# Identification of *HuSWEET* Family in Pitaya (*Hylocereus undatus*) and Key Roles of *HuSWEET12a* and *HuSWEET13d* in Sugar Accumulation

**DOI:** 10.3390/ijms241612882

**Published:** 2023-08-17

**Authors:** Rui Jiang, Liangfang Wu, Jianmei Zeng, Kamran Shah, Rong Zhang, Guibing Hu, Yonghua Qin, Zhike Zhang

**Affiliations:** Guangdong Provincial Key Laboratory of Postharvest Science of Fruits and Vegetables/Key Laboratory of Biology and Genetic Improvement of Horticultural Crops (South China), Ministry of Agriculture and Rural Affairs, College of Horticulture, South China Agricultural University, Guangzhou 510642, China; 18250650763@163.com (R.J.); 18867417851@stu.scau.edu.cn (L.W.); zjm6543212023@163.com (J.Z.); kamranshah801@scau.edu.cn (K.S.); r-zhang@scau.edu.cn (R.Z.); guibing@scau.edu.cn (G.H.)

**Keywords:** pitaya, genome-wide, *SWEET* genes family, fruit

## Abstract

The sugar composition and content of fruit have a significant impact on their flavor and taste. In pitaya, or dragon fruit, sweetness is a crucial determinant of fruit taste and consumer preference. The sugars will eventually be exported transporters (SWEETs), a novel group of sugar transporters that have various physiological functions, including phloem loading, seed filling, nectar secretion, and fruit development. However, the role of SWEETs in sugar accumulation in pitaya fruit is not yet clear. Here, we identified 19 potential members (*HuSWEET* genes) of the *SWEET* family in pitaya and analyzed their conserved motifs, physiochemical characteristics, chromosomal distribution, gene structure, and phylogenetic relationship. Seven highly conserved α-helical transmembrane domains (7-TMs) were found, and the HuSWEET proteins can be divided into three clades based on the phylogenetic analysis. Interestingly, we found two *HuSWEET* genes, *HuSWEET12a* and *HuSWEET13d*, that showed strong preferential expressions in fruits and an upward trend during fruit maturation, suggesting they have key roles in sugar accumulation in pitaya. This can be further roughly demonstrated by the fact that transgenic tomato plants overexpressing *HuSWEET12a*/*13d* accumulated high levels of sugar in the mature fruit. Together, our result provides new insights into the regulation of sugar accumulation by *SWEET* family genes in pitaya fruit, which also set a crucial basis for the further functional study of the *HuSWEETs.*

## 1. Introduction

Sugars are the predominant product of photosynthesis and play crucial roles in signal transduction, osmotic regulation, molecule transport, transient energy storage, and stress resistance during plant growth and development [1,2]. The sugars are synthesized in leaves and transported to non-photosynthesis organs (seeds, roots, and fruits) to fulfill sufficient plant growth [2,3,4]. So far, three families of eukaryotic sugar transporters have been identified, including sucrose transporters (SUTs), monosaccharide transporters (MSTs), and sugars will eventually be exported transporters (SWEETs) [1,5].

SWEETs (also known as the PQ-loop-repeat) are a new type of sugar transporters, which belong to the MtN3/saliva family (Pfam code PF03083) [6,7]. Three α-helical transmembrane domains (3-TMs) were found in the MtN3/saliva domain. Eukaryotic SWEET proteins typically comprise seven TMs that are composed of tandem repeats of two 3-TMs separated by a single TM [8]. SWEETs function as uniporters and participate in the uptake and efflux of different sugar substrates across cell membranes in pace with a concentration gradient [3,9,10]. Up to now, *SWEET* genes in many plant species have been reported, such as *Arabidopsis thaliana* [6], *Oryza sativa* [11], *Glycine max* [12], *Vitis vinifera* [13], *Citrus sinensis* [14], *Malus pumila* [15], *Lycopersicon esculentum* [16], *Litchi chinensis* [17], and *Eriobotrya japonica* [18]. In general terms, SWEET proteins are divided into four clades (clade I, clade II, clade Ⅲ, and clade IV) with distinct characteristics. SWEETs in clade I and clade II preferentially transport hexoses; members of clade Ⅲ mainly prefer sucrose; and members of clade IV are efficient fructose transporters [5,6].

The *SWEET* gene family is usually involved in different physiological processes in plants, for example, phloem loading [19], nectar secretion [20,21,22], seed filling [23,24], plant–pathogen interaction [6], biotic and abiotic stresses [25,26,27], pollen nutrition [28], and fruit development [29]. The accumulation of sugar is the core factor to determining fruit quality and yield. And, sugar transporters play integral roles in sugar partitioning and accumulation [30]. Many studies have shown that SWEET proteins were involved in sugar accumulation in fruit. In watermelon fruit, *ClSWEET3* could help hexose uptake into fruit cells from the intercellular space, and overexpression of *CISWEET3* could improve sugar contents [31]. *DlSWEET2a*/*2b*/*3a*/*16a* showed close relationships to alterations in sugar contents during longan fruit development and might be involved in fruit sugar accumulation [32]. *SlSWEET1a* participated in sugar regulation in tomato fleshy fruit [33]. In addition, *SlSWEET7a* and *SlSWEET14* were highly expressed in the fruit and silencing *SlSWEET7a* and *SlSWEET14* increased hexose contents in tomato fruit [29]. In cucumber, the hexose transporter CsSWEET7a mediated phloem unloading in companion cells for fruit development [34]. Overexpressing *VvSWEET10* in grapevine calyx and tomatoes could significantly increase hexose and total sugar contents [35]. Additionally, sucrose is the main carbon form transported from source to sink, which is important for sugar accumulation in the fruit and some SWEETs can specifically transport sucrose [36]. For example, *PuSWEET15* and *CitSWEET11d* were significantly positively correlated with sucrose concentration and promoted accumulation of sucrose in pear and citrus fruit, respectively [37,38]. These studies showed that *SWEETs* play key roles in sugar distribution and accumulation in fruit. However, no information is available about the molecular characteristics and gene functions of *SWEETs* in pitaya.

Pitaya belongs to *Hylocereus* and *Selenicereus* in the Cactaceae family of Caryophyllales. Based on the color of the peel and flesh, it can be divided into three main types: red peel with red pulp (*H. monacanthus* or *H. polyrhizus*), red peel with white pulp (*H. undatus*), and yellow peel with white pulp (*H. undatus*, *H. megalanthus,* or *S. megalanthus*) [39]. It has a delicate taste and is rich in sugars, organic acids, vitamins, betalain, and other nutrients, among which soluble sugar content is a critical factor affecting the flavor and taste quality of pitaya [39]. Pitaya is classified as a hexose accumulation fruit. The main soluble sugars in the fruit of pitaya are glucose, fructose, and sucrose. And, in mature pitaya fruit, glucose is the predominant soluble sugar. The concentration of sugar gradually increased during fruit development and reached a maximum level at the mature stage [40,41,42]. In addition, sugar accumulation varied in different sections of pitaya pulp, and the sugar content in the center section was significantly higher than that of the outside section [42]. The accumulation of soluble sugars in fruits is determined by synthesis, degradation, transport, and storage [43]. *HpINV2* and *HpSuSy1* were well correlated with the content of glucose and fructose in pitaya, and *HpWRKY3* activated the expressions of *HpINV2* and *HpSuSy1* [44]. These results showed that *HpINV2*, *HpSuSy1,* and *HpWRKY3* play important roles in sugar metabolism during fruit development. However, there is little information available about sugar transport in pitaya.

In this study, the *SWEET* family genes (*HuSWEET*) were identified across the whole pitaya genome. Expression profiles of *HuSWEETs* were analyzed in different tissue/organs and fruit development stages of the ‘Guanhuahong’ pitaya and functions of two members, *HuSWEET12a* and *HuSWEET13d*, were preliminarily validated. The present study aims to elucidate the functional roles of *HuSWEET* genes in sugar accumulation in pitaya and identify candidate genes that could contribute to fruit quality improvement in pitaya breeding programs.

## 2. Results

### 2.1. Identification and Phylogenetic Analyses of the HuSWEET Family

A total of 19 candidate *SWEET* genes with two MtN3/saliva domains were identified in the *H*. *undatus* genome [45]. These *SWEET* genes were named *HuSWEET1* to *HuSWEET13e* according to their homologous genes in *A*. *thaliana*. Multiple sequence alignments of these 19 HuSWEET proteins showed that the 7-TMs were highly conserved (Appendix A), which is consistent with results in other reported plants [46], as well as the characteristics of other plant SWEET proteins. The physicochemical characteristics of *HuSWEETs* were also analyzed, including the length of open reading frames (ORFs), the numbers of amino acids (aa), molecular weights (MW), isoelectric points (pI), hydrophilicity, and instability index (Table 1). The ORFs of the *HuSWEET* genes ranged from 708 to 1029 bp in length, encoding amino acids from 235 to 342 aa. The MW ranged from 26.30 (*HuSWEET2*) to 38.40 kDa (*HuSWEET13f*) and their pI-values ranged from 5.71 to 9.62, with an average of 8.76. The hydrophilicity of the 19 HuSWEET proteins is more than 0, indicating that these are hydrophobic proteins. Nine HuSWEET proteins (HuSWEET2/7a/7b/11/12a/12b/12d/13c/13d) have an instability index of more than 40, suggesting that these HuSWEETs are unstable proteins.

To explore the phylogenetic relationship of HuSWEETs, we downloaded SWEET proteins from pitaya, *A*. *thaliana*, rice, and grape to construct a phylogenetic tree using the maximum likelihood (ML) method. As shown in Figure 1, SWEET proteins from *A. thaliana*, rice, and grape were classified into four clades (clade I to clade IV), while the HuSWEET proteins were clustered into three clades. There is no HuSWEET member in clade Ⅳ.

### 2.2. Conserved Motif and Gene Structure Analyses of HuSWEETs

We further constructed a phylogenetic tree only using the 19 SWEET family proteins in pitaya, and classified them into three groups (Figure 2A) which is consistent with the result in Figure 1. We identified eight motifs based on the Multiple Em for Motif Elicitation (MEME) analysis (Figure 2B). There are five motifs shared by all HuSWEET proteins, indicating they are the key motifs that affect the function of the HuSWEET proteins. We also analyzed the exon–intron structures of the *HuSWEET* genes using TBtools [47], and found that *HuSWEET* genes in the same clade always exhibited similar exon–intron structure.

### 2.3. Chromosomal Localizations and Synteny Analyses of HuSWEETs

According to the location information of the pitaya genome database, the distribution of each *HuSWEET* gene on the chromosome was determined. The results showed that *HuSWEET* genes were unequally distributed on seven of the eleven chromosomes (Chrs) of pitaya (Figure 3). *HuSWEET* genes were mainly concentrated at both ends of the chromosomes; the most *HuSWEET* genes were distributed on Chr4 with seven (36.8%), followed by Chr1 with three genes (15.8%). Notably, Chr2, Chr7, Chr8, and Chr9 all contained two genes, but Chr3 had only one *HuSWEET* gene.

Gene/segmental duplication is a major mechanism driving gene family expansion with potential to create new functions. Results from synteny analyses of *HuSWEET*s showed that there was one pair of segmental duplicate events (*HuSWEET11* and *HuSWEET12d*) occurring on Chr1 and Chr4 (Figure 4).

### 2.4. Expression Profiles of HuSWEETs in Different Pitaya Tissues and Organs

To study the roles of *HuSWEET* genes in pitaya, we analyzed the expression patterns of *HuSWEET*s in different tissues and organs of ‘Guanhuahong’ pitaya (Figure 5). The results showed that most *HuSWEET* gene members expressed in all detected tissues and organs of pitaya, but the expression level varied dramatically. Most *HuSWEET* genes were strongly expressed in flowers and fruits. *HuSWEET4a*, *HuSWEET4b*, *HuSWEET10*, *HuSWEET12b*, *HuSWEET12c,* and *HuSWEET13c* had the highest expression in stamens, and *HuSWEET9* was highly expressed in styles. Comparatively, *HuSWEET7a*, *HuSWEET12a*, *HuSWEET13d*, and *HuSWEET13e* were highly expressed in pulps. In addition, *HuSWEET12a*, *HuSWEET13a*, *HuSWEET13b*, *HuSWEET13d*, and *HuSWEET13f* were strongly expressed in stems. These results indicated that *HuSWEET*s have diverse functions and *HuSWEET7a*, *HuSWEET12a*, *HuSWEET13d*, and *HuSWEET13e* are involved in sugar accumulation of pitaya

### 2.5. Changes in Soluble Sugars during Pitaya Fruit Development

To investigate the sugar accumulation pattern in pitaya fruit, we measured the soluble sugar content in pulps of ‘Guanhuahong’ pitaya during fruit development (Figure 6A,B). Total soluble sugar content increased throughout fruit development process, along with gradual increases in glucose, fructose, and sucrose levels. Glucose and fructose contents in pulp showed rapid accumulation from 23 days after flowering (DAF) to 25 DAF, reaching maximum levels at full maturation (35 DAF). At this stage, glucose and fructose contents were 57.97 mg/g and 36.71 mg/g, respectively, significantly higher than sucrose levels (19.74 mg/g). These findings suggest that glucose is the main soluble sugar in mature pitaya fruit, consistent with previous studies [40,42].

### 2.6. Expression Analyses of HuSWEETs in Pitaya Fruits

To explore the relationship between *HuSWEET* genes and fruit sugar accumulation, we further analyzed the expression patterns of *HuSWEETs* in the pulps at different fruit developmental stages of ‘Guanhuahong’ pitaya. As shown in Figure 7A, *HuSWEET1*, *HuSWEET9*, *HuSWEET10,* and *HuSWEET13a* showed irregular expression patterns during fruit development of ‘Guanhuahong’ pitaya. Ten *HuSWEET* genes (*HuSWEET2*, *HuSWEET4b*, *HuSWEET7a*, *HuSWEET7b*, *HuSWEET12b*, *HuSWEET12c*, *HuSWEET13b*, *HuSWEET13c*, *HuSWEET13e,* and *HuSWEET13f*) exhibited gradually down-regulated expression patterns in pulps during fruit development of ‘Guanhuahong’ pitaya; *HuSWEET7a*, *HuSWEET7b,* and *HuSWEET12b* showed higher expression level and change among the ten *HuSWEETs*. On the contrary, the expression levels of *HuSWEET4a*, *HuSWEET11, HuSWEET12a*, *HuSWEET12d*, and *HuSWEET13d* showed up-regulated trends. Notably, *HuSWEET4a* had the highest expression level at 32 DAF while *HuSWEET11*/*12d* and *HuSWEET12a*/*13d* reached the highest expression at 27 and 35 DAF, respectively. Furthermore, higher expression levels of *HuSWEET12a*, *HuSWEET12d*, and *HuSWEET13d* were detected in the pulps of mature fruit than those of the other *HuSWEET*s (Figure 7B).

Additionally, we also investigated the correlation between the expression of *HuSWEETs* and the content of main sugars during fruit development of ‘Guanhuahong’ pitaya. As shown in Table 2, the expression pattern of most *HuSWEETs* exhibited negative correlations with the contents of major sugars in pitaya fruit. Specifically, *HuSWEET12c* and *HuSWEET13f* showed significantly negative correlations with total soluble sugar, sucrose, glucose, and fructose. On the other hand, the expressions of five *HuSWEETs* (*HuSWEET4a*, *HuSWEET11, HuSWEET12a*, *HuSWEET12d,* and *HuSWEET13d*) showed positive correlations with the contents of major sugars in pitaya fruit. Of them, the expressions of *HuSWEET12a* and *HuSWEET13d* showed significant and positive correlation with the accumulation of total soluble sugar, sucrose, glucose, and fructose during fruit maturation. Results from gene expressions and sugar contents suggested that *HuSWEET12a* and *HuSWEET13d* are likely contributed to sugar accumulation during the fruit development of pitaya and were selected as the candidate genes for further analyses.

### 2.7. Plasma Membrane-Localized HuSWEET12a and HuSWEET13d Proteins

To further investigate the potential functions of *HuSWEET12a* and *HuSWEET13d* in the accumulation of soluble sugars in the pitaya fruit, we analyzed their subcellular localization (Figure 8). We first constructed 35S-HuSWEET12a-GFP and 35S-HuSWEET13d-GFP vectors and then co-expressed them with mCherry-labeled plasma membrane marker protein (pCAM35S::At5g19750-mCherry) in *N. benthamiana* leaves. The fluorescence of HuSWEET12a-GFP and HuSWEET13d-GFP was found to overlap with the red fluorescence of PM-mCherry, suggesting that *HuSWEET12a* and *HuSWEET13d* are localized in the plasma membrane. 

### 2.8. Overexpression of HuSWEET12a and HuSWEET13d Could Increase Sugar Contents of Tomato Fruits

To further study the function of *HuSWEET12a* and *HuSWEET13d*, they were transferred into the ‘Micro-Tom’ tomato (*Solanum lycopersicum*) cotyledons using an *Agrobacterium*-mediated method, and the transformed plants were selected based on kanamycin resistance. Two independent T_1_ generation lines were obtained for further analyses with overexpression of either *HuSWEET12a* (12a-OE-4/11) or *HuSWEET13d* (13d-OE-24/29). There were no significant differences in morphology between the transgenic lines and the wild-type (WT) plants (Figure 9A,B). Transcriptome analysis confirmed successful expression of *HuSWEET12a* and *HuSWEET13d* in tomatoes (Figure 9C,D). At 50 DAF, the total sugar, glucose, and fructose contents in *HuSWEET12a*-OE-4/11 and *HuSWEET13d*-OE-24/29 lines were significantly higher than those in WT lines. In addition, sucrose content was also significantly increased in the *HuSWEET12a*-OE-4 and *HuSWEET13d*-OE-29 lines compared with WT (Figure 9E,F).

## 3. Discussion

Exported transporters or SWEETs can mediate the uptake and efflux of sugars, which play critical roles in plants [6]. Genome-wide identification of *SWEET* genes has been widely studied in many plants, and the *SWEET* gene members vary from 7 to 108 [6,13,48,49]. In different plants, the number of *SWEET* genes is various, which may be caused by tandem or segmental duplication events [12,48,50]. In our study, a total of 19 *SWEET* genes were identified in pitaya, and one segmentally duplicated event (*HuSWEET11* and *HuSWEET12d*) was identified in pitaya. The number of *SWEET* genes in pitaya is similar to that of *A*. *thaliana* [6] and litchi [17]. The length of 19 HuSWEET proteins ranged from 235 aa to 352 aa, similar to cucumber [40], banana [51], and litchi [17]. HuSWEET proteins are similar in structure to SWEET proteins in most other plants which contain seven TMs including two typical MtN3 domains. Gene structure analyses showed that most *HuSWEET* genes had six exons; the similar results were also found in cucumber [52], litchi [17], and apple [53]. In general, the acquisition or loss of introns and the distribution pattern of exon–intron may lead to the complexity of gene structure, thereby obtaining new gene functions, which are considered key factors affecting the evolutionary mechanism of gene families [54]. *HuSWEET* genes of the same clade shared similar conserved motif composition and exon–intron structure; these results suggested that the *SWEET* gene family in pitaya is relatively conservative in the evolutionary process.

*SWEET* genes have different expression patterns in different plant organs and tissues, which are involved in a variety of physiological processes [55]. In our study, most of the *HuSWEET* genes were expressed in the all-detected tissues and organs with different expression patterns. Jeena et al. [55] found that the *SWEET* gene family plays an important role in the reproductive development of plants. In our study, all *HuSWEET* genes were expressed in flowers, similar to the results in *A. thaliana* [6,22,56], *M. domestica* [53], *V. vinifera* [13], and *Jasminum sambac* [57]. Additionally, *HuSWEET9* was highly expressed in the style and ovary, and its orthologous gene, *AtSWEET9*, was involved in nectar secretion in *A*. *thaliana* [22]. The results suggested that *HuSWEET9* may be responsible for nectar secretion in pitaya. In *A*. *thaliana, AtSWEET2* was abundantly expressed in roots and involved in the transfer of sucrose from the root tip epidermal cells to the root margins. The highest level of *HuSWEET2* was detected in roots compared with other tissues and organs of pitaya, which was similar to the expression pattern of *AtSWEET2*. Studies have shown that *SWEETs* can regulate sugar partitioning and accumulation in leaves and fruits of plants [58]. In tomato, *SlSWEET1a* was highly expressed in leaves, and *SlSWEET7a* and *SlSWEET14* were highly expressed in fruit, which played key roles in the sugar accumulation of leaves and fruit, respectively [29,33]. In pitaya, *HuSWEET12a*, *HuSWEET13a*, *HuSWEET13b*, *HuSWEET13d*, *HuSWEET13e,* and *HuSWEET13f* were highly expressed in the stem or mature fruit, suggesting that they may play important roles in sugar accumulation in the stem or mature fruit of pitaya.

Sugar content is a major factor of fruit quality [32], and the functional identification of SWEET in sugar accumulation of fruit has been extensively studied. In apple, *MdSWEET9b* and *MdSWEET15a* were significantly correlated to fruit sugar contents and may be involved in sugar accumulation [59]. Notably, *PuSWEET15* showed positive correlation with sucrose content, and overexpression of *PuSWEET15* in pear fruit increased significantly sucrose levels [38]. In this study, we found that the contents of soluble sugars increased during pitaya fruit development; expression levels of *HuSWEET7a*, *HuSWEET7b*, *HuSWEET12b*, *HuSWEET*13c, and *HuSWEET13e* showed strong down-regulated expression patterns during fruit development. These results suggested that *HuSWEET7a*, *HuSWEET7b*, *HuSWEET12b*, *HuSWEET13c,* and *HuSWEET13e* may play roles in sugar accumulation in the early stages of pitaya fruit development, and similar results were also reported for pineapple [60]. In addition, higher expression levels of *HuSWEET12a*, *HuSWEET12d,* and *HuSWEET13d* were detected in mature pitaya fruits. Furthermore, *HuSWEET12a* and *HuSWEET13d* were significantly and positively correlated with the accumulation of soluble sugar in pitaya fruits. Compared with WT, overexpression of *HuSWEET12a* and *HuSWEET13d* in tomato resulted in higher contents of sucrose, glucose, fructose, and total sugar. These results suggested that *HuSWEET12a* and *HuSWEET13d* are possibly involved in the sugar accumulation of pitaya.

## 4. Materials and Methods

### 4.1. Plant Materials

‘Guanhuahong’ pitaya (*H. monacanthus*) and ‘Micro-Tom’ tomato (*Solanum lycopersicum)* were selected to be materials in this study. ‘Guanhuahong’ is an excellent pitaya variety with red peel and red pulp which were selected by the Dongguan Institute of Forestry Science and South China Agricultural University (SCAU, Guangzhou, China) [61]. ‘Guanhuahong’ were grown in the orchard of SCAU, the stem, root, flower (petal, stamen, style, and ovary), fruit (peel and pulp), and fruits from the 17th, 19th, 23rd, 25th, 27th, 32nd, and 35th (DAF) of pitaya were collected for gene expression. All samples were frozen in liquid nitrogen immediately and kept at −80 °C for further study.

‘Micro-Tom’ tomatoes were grown in a plastic container (9 cm × 9 cm) with nutritious soil (Jiffy), and cultivated in the greenhouse (16 h/8 h (light/dark); a temperature of 25 ± 1 °C and relative humidity of 65%) were used for genetic transformation.

### 4.2. Measurement of Soluble Sugars

The soluble sugar (including fructose, glucose, and sucrose) contents were measured according to the method of Zhang et al. [42] with minor modifications. A total of 0.5 g of samples were used to extract soluble sugars with 4 mL and 2 mL of 90% alcohol (*v*/*v*), respectively. The supernatants were dried using Concentrator plus (Eppendorf AG, Hamburg, Germany). C18 column and 0.45 µm filters were used to filter supernatants. Samples were determined in an Agilent 1200LC HPLC system (Agilent Technologies, Waldbronn, Germany). The content of total soluble sugar was measured according to Xie et al. [41].

### 4.3. Identification of HuSWEET Family Genes in Pitaya

The HMMER profile [62] of the MtN3/saliva motif (PF03083) was downloaded from PFAM (https://pfam.xfam.org/) (accessed on 20 October 2022) [63] and was used as a query to identify *SWEET* genes from the pitaya genome database (http://www.pitayagenomic.com) (accessed on 20 October 2022) [45]. Putative *SWEET* family genes were then confirmed through PFAM and SMART with an e-value less than 1 × 10^−5^ (https://smart.embl-heidelberg.de/) (accessed on 20 October 2022) to eliminate genes without the reported conserved domains and motifs of the *MtN3*/*saliva*/*SWEET* family members. The final *SWEET* gene family members were identified using the two MtN3/saliva conserved domains. ExPASy (http://web.expasy.org/potparam/) (accessed on 20 October 2022) was used to predict the characteristic of the pitaya SWEET proteins (the numbers of amino acids, MW, pI, hydrophilicity, and stability).

### 4.4. Gene Structure and Conserved Motifs Analyses of HuSWEETs

The exon/intron organizations of *HuSWEET*s were drawn by TBtools software (v 1.098769) [47]. The MEME (https://meme-suite.org/) (accessed on 20 October 2022) was used to analyze the conserved motifs. The optimized parameters of MEME were set up as follows: the maximum number was eight identified motifs and the optimum width was from 1 to 50 residues of each motif. DNAMan software (v 6.0.3.99) was used to obtain the multiple sequence alignments of the amino acid sequences.

### 4.5. Locations in the Chromosome, Phylogenetic, and Duplication Pattern of HuSWEETs

We downloaded the complete amino acid sequences of *SWEETs* for the phylogenetic analysis from the Arabidopsis Information Resource (TAIR) (https://www.arabidopsis.org/) (accessed on 20 October 2022) and NCBI databases (https://www.ncbi.nlm.nih.gov) (accessed on 20 October 2022), respectively. The phylogenetic trees of pitaya, grape, rice, and *A. thaliana* SWEET proteins were generated using the ML method in MEGA Ⅹ (v 10.1.6) with 1000 bootstrap replicates [46]. The sequences were listed in Appendix A. The EVOLVIEW online tool (https://evolgenius.info//evolview-v2/#login) (accessed on 20 October 2022) was used to exhibit all the evolutionary trees [64]. The chromosomal locations of *HuSWEET* genes were identified from the pitaya genome database. The MapChart software (v 1.0.0.0) was used to draw the locations of *HuSWEETs*. TBtools software was used for the synteny analyses of each *HuSWEET*.

### 4.6. RNA Isolation and Gene Expression Analyses

The EASYspin Plus Complex Plant RNA Kit (RN53) (Aidlab Biotechnology, Beijing, China) was used for the RNA isolation. The PrimeScript™ RT Reagent Kit with gDNA Eraser (TaKaRa, Shiga, Japan) was used to synthesize the single-stranded cDNA. Quantitative real-time PCR (qRT-PCR) was performed using an CFX384-Real-Time System (C1000 Touch Thermal Cycler, Bio-Rad, Irvine, CA, USA) with the RealUniversal Color PreMix (SYBR Green) (TIANGEN, Beijing, China). The specific primers for qRT-PCR were designed at NCBI (https://www.ncbi.nlm.nih.gov/tools/primer-blast/) (accessed on 20 October 2022). *Actin* was selected to be the internal control for gene expression analyses [65]. The primers were listed in Appendix A. All experiments were conducted in three biological replicates. The comparative 2^−ΔΔC^_T_ method was used to calculate the relative expression levels of each transcript [66].

### 4.7. Subcellular Localization

The coding regions of the *HuSWEET12a*/*13d* genes (without stop codons) with *Xba* I and *Kpn* I restriction sites were cloned into the pC18-35S:GFP vector. The primers are listed in Appendix A. The recombinants were induced into *Agrobacterium tumefaciens* strain GV3101 (pSoup-p19) and used for transient expression in *N. benthamiana*. The infected *N. benthamiana* leaves were examined 48 h after infiltration. Fluorescence microscopes (ZEISS LCM-800, Oberkochen, Germany) were used to capture the GFP signal. All assays were repeated three times.

### 4.8. Genetic Transformation

The coding sequences of *HuSWEET12a*/*13d* were cloned into the pPZP6K90 vector and transformed into *A. tumefaciens* strain GV3101. Cotyledons from seven- to ten-day-old seedlings of ‘Micro-Tom’ tomato were selected as explants. Both ends of each cotyledon were removed to allow it to absorb the bacterial suspension. The explants were dipped in the bacterial suspension (OD_600_ = 0.5–0.6) for 20 min and blotted dry on a sterilized paper towel. Then, the explants were placed in co-culture medium (MS medium with 30 g/L sucrose, 6 g/L agar, 2.0 mg/L 6-BA, and 0.2 mg/L IAA, pH 5.4) and incubated in darkness for two days at 24 °C. After co-culture, the explants were transferred to screening medium (30 g/L sucrose, 6 g/L agar, 2.0 mg/L 6-BA, 0.2 mg/L IAA, 100 mg/L kanamycin, and 300 mg/L timentin, pH 5.8). When calli with adventitious buds developed from the explants, the cotyledons were cut off and transferred to a new screening medium with 50 mg/L kanamycin. Buds higher than one cm were cut and transferred to rooting medium (MS medium with 30 g/L sucrose, 6 g/L agar, 0.5 mg/L IBA, 25 mg/L kanamycin and 150 mg/L timentin, pH 5.8). The transformed ‘Micro-Tom’ tomato plantlets were analyzed using PCR and qRT-PCR. The primers were listed in Appendix A. Fruits from the T_1_ generation of overexpressing *HuSWEET12a* and *HuSWEET13d* lines were used for sugar concentration analyses.

## 5. Conclusions

In summary, nineteen *HuSWEET* genes were identified in pitaya and distributed unevenly across seven of the eleven chromosomes. *HuSWEET12a* and *HuSWEET13d* were found to be significantly and positively correlated with the accumulation of total soluble sugar, sucrose, glucose, and fructose during pitaya fruit maturation. *HuSWEET12a* and *HuSWEET13d* were located on the plasma membrane and overexpressing them in tomato could promote sugar accumulation. These results suggest that *HuSWEET12a* and *HuSWEET13d* are possibly involved in the sugar accumulation of pitaya.

## Figures and Tables

**Figure 1 ijms-24-12882-f001:**
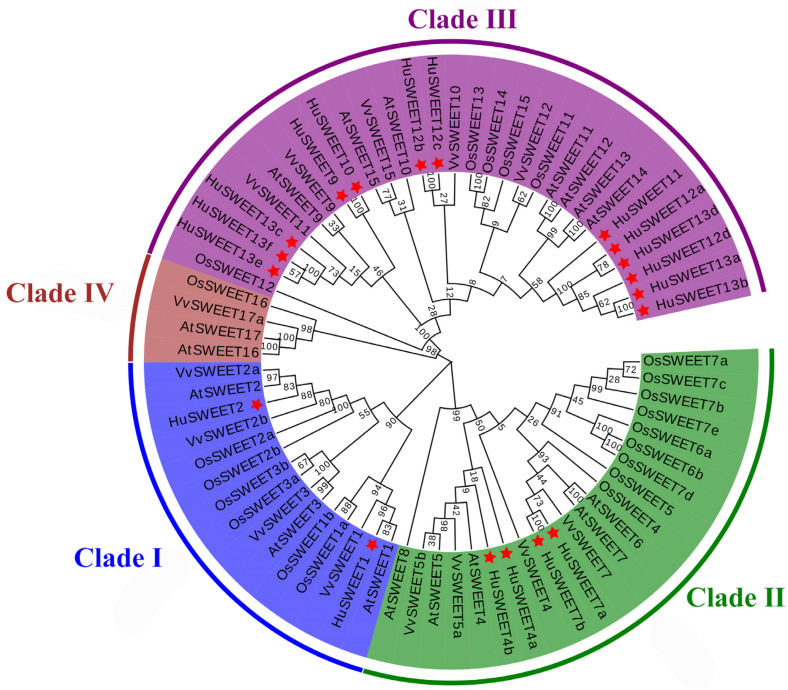
The maximum likelihood phylogenetic tree of SWEET proteins from pitaya and *A*. *thaliana*, rice, and grape. Four clades labeled with different colors. HuSWEETs are indicated by a red asterisk.

**Figure 2 ijms-24-12882-f002:**
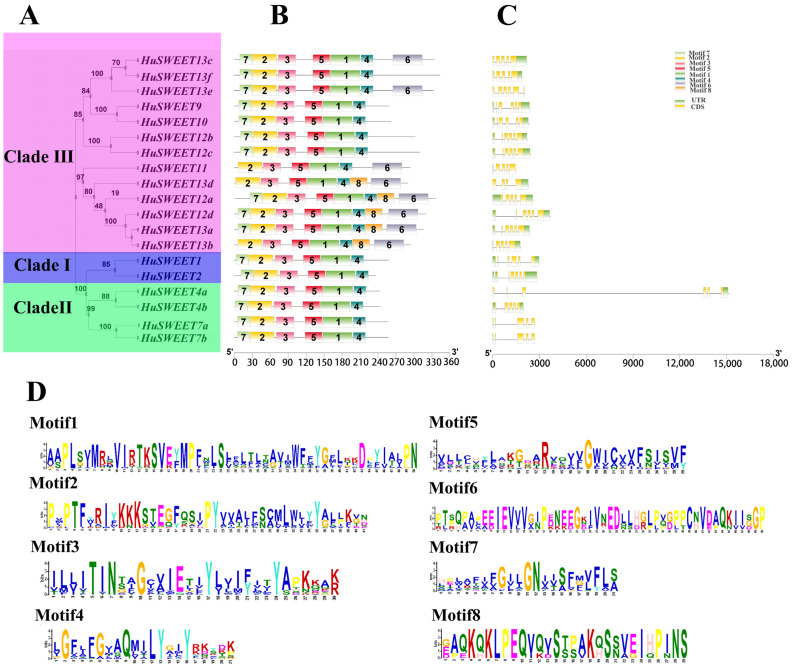
Phylogenetic relationship, conserved motif, and gene structure analyses of *HuSWEET* genes. (**A**) The neighbor-joining phylogenetic tree of *HuSWEETs* was generated using MEGA Ⅹ with 1000 bootstrap replicates. (**B**) The motif compositions of HuSWEET proteins. Eight motifs were displayed in rectangles of different colors. (**C**) Exon–intron structure of *HuSWEET* genes. Yellow rectangles indicate exons, black lines indicate introns, and green rectangles represent the UTR region. (**D**) Amino acid sequences of the eight motifs of HuSWEET proteins.

**Figure 3 ijms-24-12882-f003:**
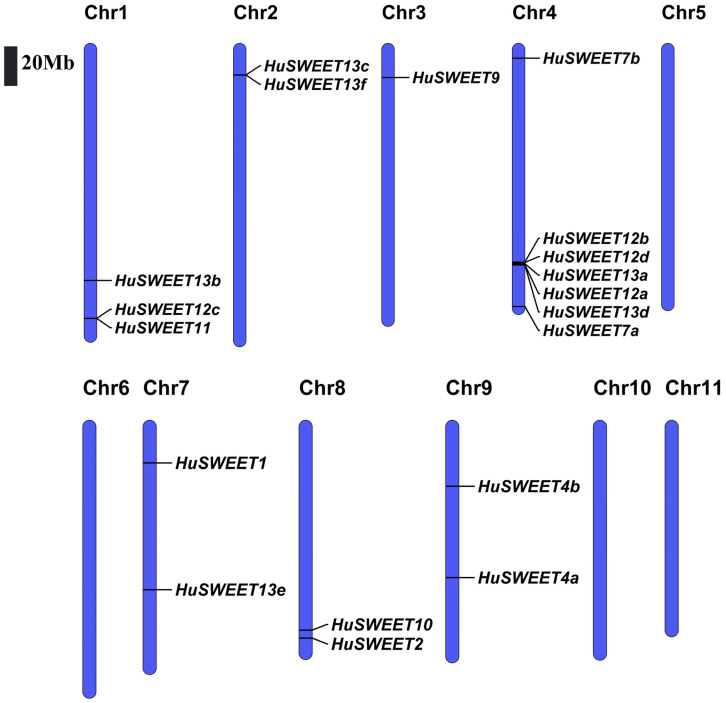
Distribution of *HuSWEET* members on the eleven chromosomes of pitaya, Bars = 20 Mb.

**Figure 4 ijms-24-12882-f004:**
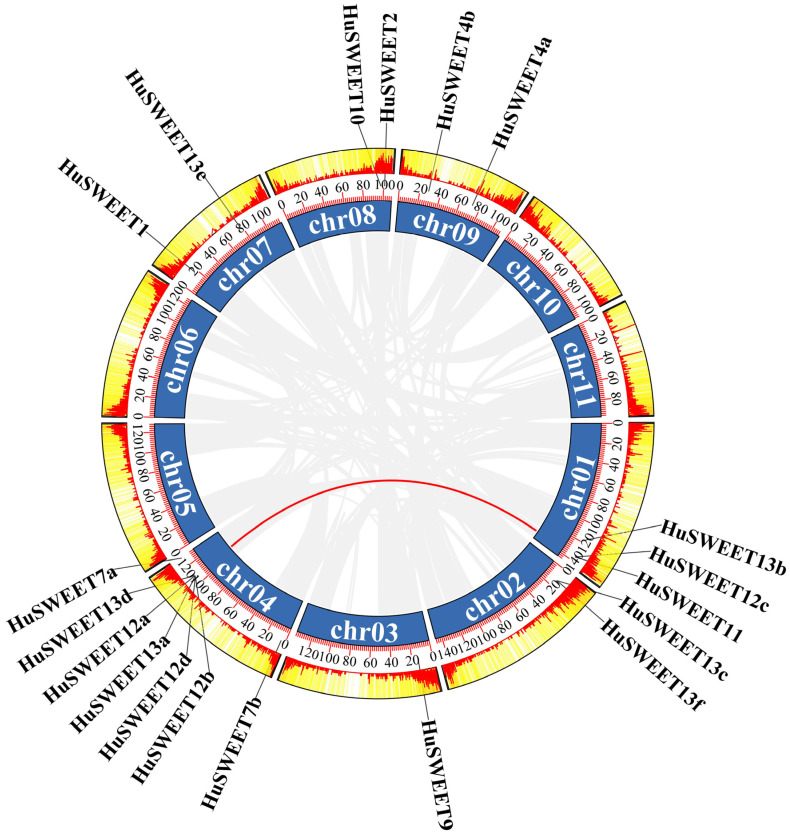
Synteny analyses of *HuSWEETs*. Gray lines represent all synteny blocks in the pitaya genome, and the red lines indicate the duplicated *HuSWEET* gene pair.

**Figure 5 ijms-24-12882-f005:**
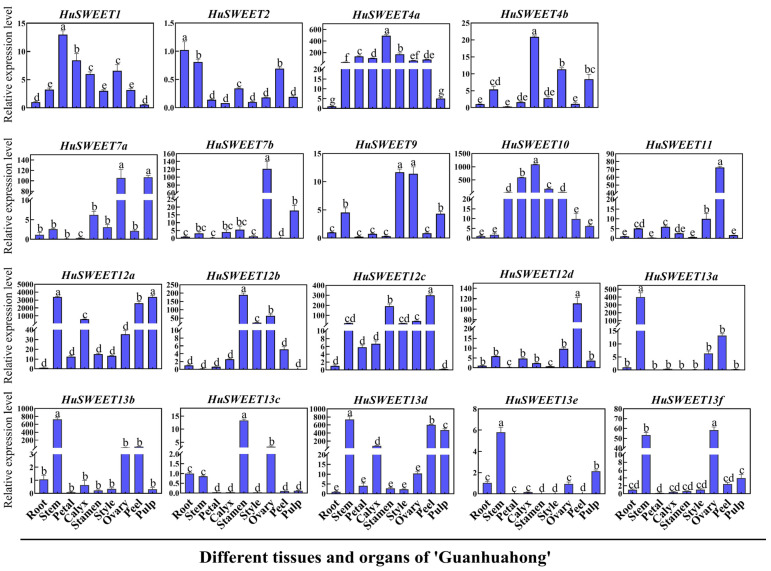
Expression analyses of 19 *HuSWEET* genes in different pitaya tissues and organs. The data are shown as the means ± SDs of three independently biological replicates. The different letters above bars indicate significant differences at the *p* < 0.05 level according to Duncan’s multiple comparison tests.

**Figure 6 ijms-24-12882-f006:**
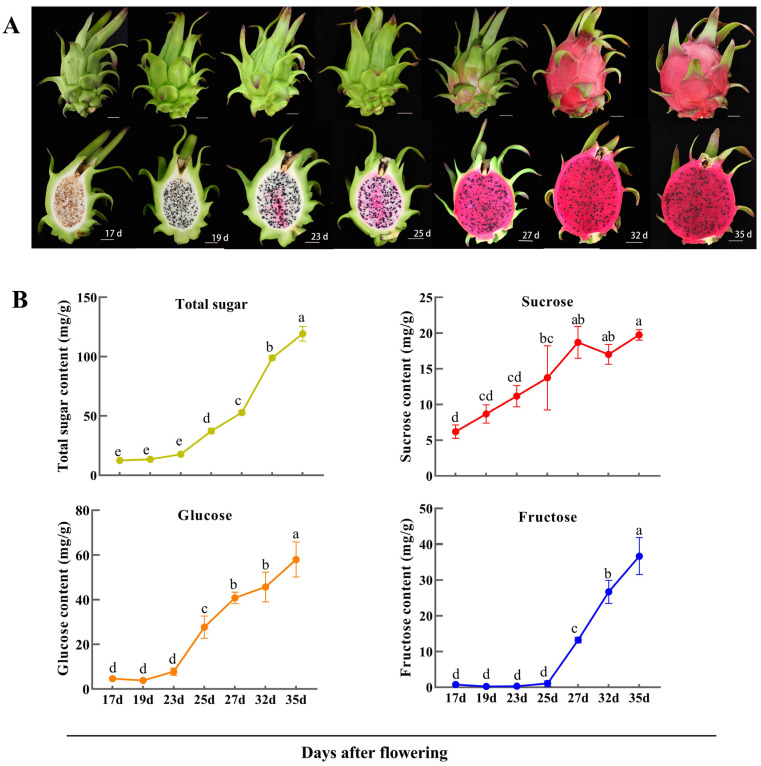
Changes in sugar contents in pulp during fruit development of pitaya. (**A**) Pitaya fruit (whole fruit and longitudinal sections) at seven developmental stages. (**B**) Contents of total sugar, glucose, fructose, and sucrose in pulps during fruit development. The data were shown as the means ± SDs of three independently biological replicates. The different letters above bars indicate significant differences at the *p* < 0.05 level according to Duncan’s multiple comparison tests.

**Figure 7 ijms-24-12882-f007:**
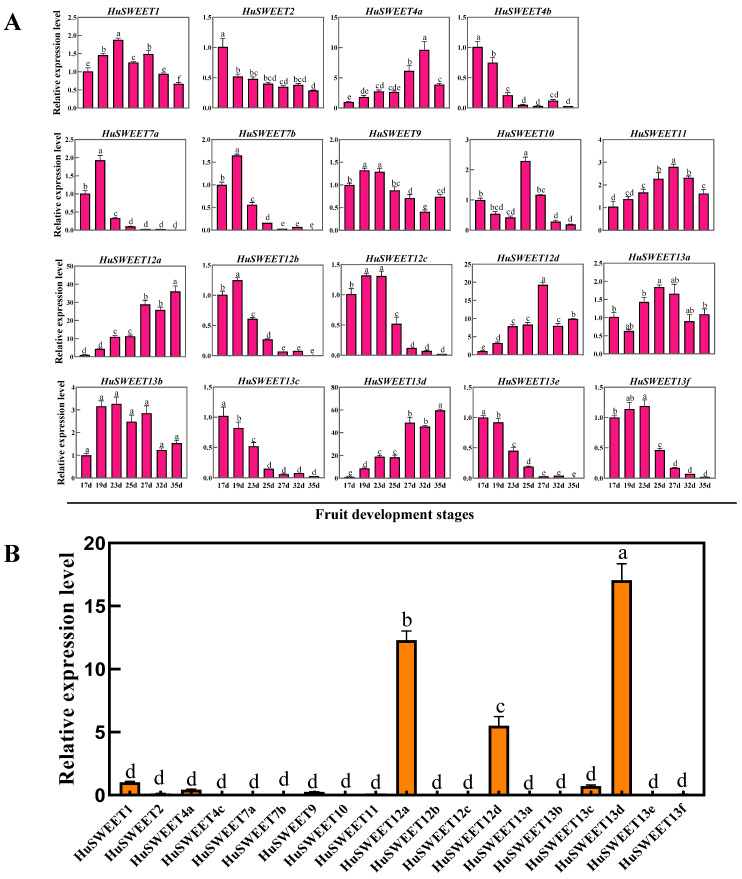
Expression patterns of *HuSWEETs* in pitaya pulps. (**A**) Expression analyses of 19 *HuSWEET* genes in pitaya pulps during fruit development. (**B**) RT-qPCR analyses of 19 *HuSWEET*s in mature fruits of pitaya. The data were shown as the means ± SDs of three independently biological replicates. The different letters above bars indicate significant differences at the *p* < 0.05 level according to Duncan’s multiple comparison tests.

**Figure 8 ijms-24-12882-f008:**
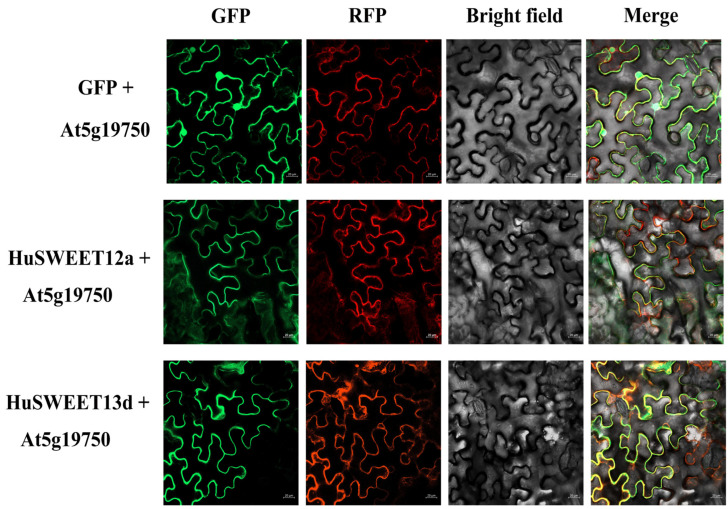
Subcellular localization of *HuSWEET12a* and *HuSWEET13d* in *N. benthamiana* leaves. Notably, At5g19750-RFP was the plasma membrane marker. Bar = 20 μm.

**Figure 9 ijms-24-12882-f009:**
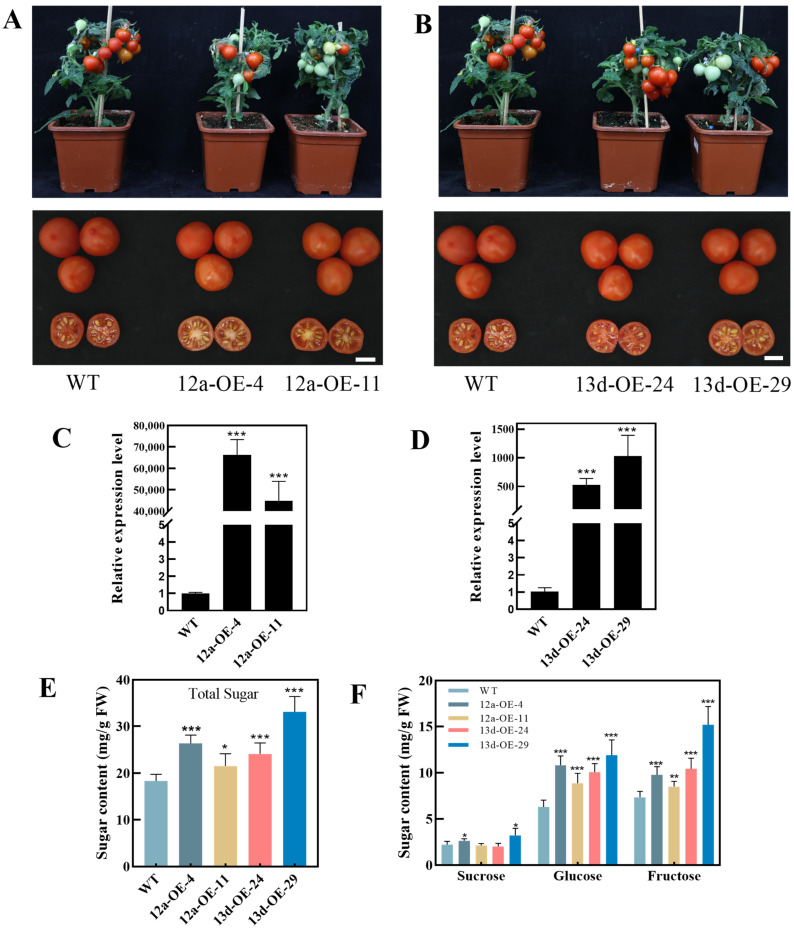
Analysis of sugars content and morphological phenotypes in *HuSWEET12a*/*13d* transgenic lines (OE) and the wild-type (WT) tomato plants. (**A**,**B**) Morphological phenotypes of *HuSWEET12a*-OE and *HuSWEET13d*-OE (Bar = 1 cm). (**C**,**D**) Transcript levels of *HuSWEET12a*/*13d* in OE and WT plants. (**E**) The content of total sugar in mature fruit of OE and WT plants. (**F**) The contents of sucrose, glucose, and fructose in mature fruit of OE and WT tomato plants. The data represent mean values ± SDs. The asterisks indicate *p*-values (* *p* < 0.05; ** *p* < 0.01, and *** *p* < 0.001) according to Student’s *t* test.

**Table 1 ijms-24-12882-t001:** Physiochemical characteristics of the *HuSWEETs* in pitaya.

Gene ID	Gene Names	Length of Open Reading Frames (bp)	No. Amino Acids (aa)	Isoelectric Points	Molecular Weights (kDa)	Hydrophilicity	Instability Index
HU07G00905.1	*HuSWEET1*	774	257	9.32	28.01	0.477	27.48
HU08G01881.1	*HuSWEET2*	708	235	9.17	26.30	0.87	56.04
HU09G00864.1	*HuSWEET4a*	729	242	9.62	27.04	0.555	36.88
HU09G00626.1	*HuSWEET4b*	732	243	9.32	27.03	0.53	28.03
HU04G02229.1	*HuSWEET7a*	771	256	9.24	27.83	0.783	44.92
HU04G00351.1	*HuSWEET7b*	771	256	9.28	27.86	0.788	46.82
HU03G01299.1	*HuSWEET9*	774	257	9.47	28.33	0.557	35.88
HU08G01549.1	*HuSWEET10*	783	260	9.32	28.57	0.635	36.59
HU01G02213.1	*HuSWEET11*	882	293	9.12	33.12	0.34	48.54
HU04G01326.1	*HuSWEET12a*	1008	335	7.02	37.37	0.542	41.98
HU04G01322.1	*HuSWEET12b*	906	301	9.14	34.13	0.772	41.03
HU01G02212.1	*HuSWEET12c*	930	309	9.45	34.74	0.593	39.71
HU04G01323.1	*HuSWEET12d*	957	318	8.74	35.53	0.501	42.51
HU04G01325.1	*HuSWEET13a*	948	315	8.96	34.87	0.534	37.13
HU01G01645.1	*HuSWEET13b*	885	294	8.85	32.44	0.486	38.55
HU02G01006.1	*HuSWEET13c*	1002	333	5.71	37.57	0.368	107.12
HU04G01327.1	*HuSWEET13d*	870	289	8.32	32.14	0.391	46.8
HU07G01376.1	*HuSWEET13e*	999	332	8.18	37.39	0.387	39.42
HU02G01007.1	*HuSWEET13f*	1029	342	8.15	38.39	0.356	36.9

**Table 2 ijms-24-12882-t002:** Correlation analyses of expression levels of *HuSWEETs* and sugar content during fruit development.

Genes	Soluble Sugar	Glucose	Fructose	Sucrose
*HuSWEET1*	−0.689	−0.598	−0.709	−0.332
*HuSWEET2*	−0.587	−0.701	−0.545	−0.840 *
*HuSWEET4a*	0.684	0.689	0.637	0.704
*HuSWEET4c*	−0.583	−0.763	−0.532	−0.885 **
*HuSWEET7a*	−0.591	−0.75	−0.549	−0.775 *
*HuSWEET7b*	−0.668	−0.838 *	−0.629	−0.851 *
*HuSWEET9*	−0.771 *	−0.839 *	−0.741	−0.729
*HuSWEET10*	−0.462	−0.161	−0.497	−0.164
*HuSWEET11*	0.322	0.588	0.267	0.728
*HuSWEET12a*	0.905 **	0.955 **	0.898 **	0.972 **
*HuSWEET12b*	−0.762 *	−0.915 **	−0.725	−0.934 **
*HuSWEET12c*	−0.839 *	−0.961 **	−0.821 *	−0.891 **
*HuSWEET12d*	0.387	0.63	0.371	0.803 *
*HuSWEET13a*	−0.089	0.258	−0.144	0.345
*HuSWEET13b*	−0.411	−0.333	−0.432	−0.097
*HuSWEET13c*	−0.727	−0.898 **	−0.682	−0.959 **
*HuSWEET13d*	0.908 **	0.951 **	0.900 **	0.972 **
*HuSWEET13e*	−0.739	−0.897 **	−0.695	−0.957 **
*HuSWEET13f*	−0.854 *	−0.973 **	−0.834 *	−0.910 **

Significant and extremely significant differences are represented by * (*p* < 0.05) and ** (*p* < 0.01), respectively.

## Data Availability

Data are contained within the article and Appendix A.

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
