# Peer review of "Identification of HuSWEET Family in Pitaya (Hylocereus undatus) and Key Roles of HuSWEET12a and HuSWEET13d in Sugar Accumulation"

_ijms, 2023, doi:10.3390/ijms241612882_

Round 1
Reviewer 1 Report
Manuscript Identification of HuSWEET family in pitaya (Hylocereus undatus) and key roles of HuSWEET12a and HuSWEET13d in
sugar accumulations by Rui Jiang et al. considers the issues of fine regulation of sugars with the prospect of using it to modify other plants.
The article is of particular interest for this branch of research.
Manuscript is not formatted according to the rules of the magazine. Literature in the text is placed in the form of the names of the first author. References not numbered
‘Guanhuahong’ pitaya (H. monacanthus) and ‘Micro-Tom’ tomato.
For tomato, soil and container sizes should be indicated or a ruler in Figure 8.
In figure 4, as well as in the materials and methods and the text of the article, there is a serious methodological error. Roots, stems, petals, ovules and all of the above are not tissue. Dear authors, consult with experts or look at the definition ... The terms mentioned refer to organs and contain many types of tissues. Be careful and correct the entire manuscript. This is unacceptable and misleading.
In general, the work contains interesting material, but should be corrected.
Author Response
1. Manuscript is not formatted according to the rules of the magazine. Literature in the text is placed in the form of the names of the first author. References not numbered.
Our response: Thanks for your suggestion. We have revised the manuscript format.
2. ‘Guanhuahong’ pitaya ( H. monacanthus) and ‘Micro-Tom’ tomato.
For tomato, soil and container sizes should be indicated or a ruler in Figure 8.
Our response: We have added the soil of the tomato and the container size in the revised manuscript.
3. In figure 4, as well as in the materials and methods and the text of the article, there is a serious methodological error. Roots, stems, petals, ovules and all of the above are not tissue. Dear authors, consult with experts or look at the definition ... The terms mentioned refer to organs and contain many types of tissues. Be careful and correct the entire manuscript. This is unacceptable and misleading.
Our response: We have revised it into “different parts (tissues or organs)” in the revised manuscript.
Reviewer 2 Report
The manuscript “Identification of HuSWEET family in pitaya (Hylocereus undatus) and key roles of HuSWEET12a and HuSWEET13d in sugar accumulation” by Jiang et al submitted to IJMS deals with important topic like identification and characterization of important gene family such as SWEET in important fruit (pitaya) and further molecular characterization by genetic transformation of tomato.
The manuscript will be of high interest to the scientific community working to that area of sugar accumulation in fruits. Below is the evaluation report.
Introduction:
The introduction is well written and point the main problems related to the topic.
Results:
Results are clearly presented.
Discussion
This part is well written.
Materials and methods
M&Ms are somehow well written.
Overall, the manuscript deserves to be published.
Author Response
Thanks for your review.
Reviewer 3 Report
Manuscript (ijms-2501724) “Identification of HuSWEET family in pitaya (Hylocereus undatus) and key roles of HuSWEET12a and HuSWEET13d in sugar accumulation” by Jiang et al. presents the results of the molecular analysis of related sugar accumulation genes in pitaya.
In general, the manuscript is well written although some revisions are required before publication. Most important point for the revision of the manuscript are:
In the whole manuscript the scientific name of the species and the name of the genes must be in italic.
Quality of Figures 2, 3 and 4 should be improved increasing font size.
Plant material used in the study must be clearly described. It is necessary to indicate the origin and pedigree of the assayed pitaya genotypes and main phenological characteristics. In addition, if there is any previous reference describing this material should also be included.
The evaluation of sugars should be indicated in the material and Method sections after the description of the plant material.
In silico search of the HuSWEET family genes in pitaya must also be clarified.
qRT-PCR analysis should be also clarified indicating the nature of the assayed technical and biological samples. In addition, the selection of the candidate genes must be justified to together with the statistical analysis used.
Genetic transformation protocol must also be completed.
Discussion and Conclusion sections should be after Result section and before material and Method section. In addition, Discussion section must be completed because at this moment is very poor.
English grammar and expression should be revised
Author Response
1. In general, the manuscript is well written although some revisions are required before publication. Most important point for the revision of the manuscript are:
In the whole manuscript the scientific name of the species and the name of the genes must be in italic.
Our response: We have revised all the scientific name of the species and the name of the genes in italic format.
2. Quality of Figures 2, 3 and 4 should be improved increasing font size.
Our response: We have increased the font size in Figure 2, 3 and 4.
3. Plant material used in the study must be clearly described. It is necessary to indicate the origin and pedigree of the assayed pitaya genotypes and main phenological characteristics. In addition, if there is any previous reference describing this material should also be included.
Our response:We have added the introduction of ‘Guanhuahong’ pitaya and the reference.
4. The evaluation of sugars should be indicated in the material and Method sections after the description of the plant material.
Our response: In the revised manuscript, we have put the evaluation of sugar part after the description of the plant material.
5. In silico search of the HuSWEET family genes in pitaya must also be clarified.
Our response: In the revised manuscript, we have added the process of the silico search of the HuSWEET genes in pitaya.
6. qRT-PCR analysis should be also clarified indicating the nature of the assayed technical and biological samples.
Our response: We have clarified that sample is biological repetition in the method sections 4.6.
7. In addition, the selection of the candidate genes must be justified to together with the statistical analysis used.
Our response: We have added the chosen of candidate genes with the statistical analysis.
8. Genetic transformation protocol must also be completed.
Our response: Genetic transformation protocol has been provided in the revised manuscript.
9. Discussion and Conclusion sections should be after Result section and before material and Method section. In addition, Discussion section must be completed because at this moment is very poor.
Our response: We have revised the format and the Discussion section according to the IJMS.
10. Comments on the Quality of English Language
English grammar and expression should be revised
Our response: We have revised the English grammar and expression throughout the manuscript.
Round 2
Reviewer 3 Report
Authors have revised corectly the manuscript